# Forms and Methods for Interferon’s Encapsulation

**DOI:** 10.3390/pharmaceutics13101533

**Published:** 2021-09-22

**Authors:** Thelvia I. Ramos, Carlos A. Villacis-Aguirre, Nelson Santiago Vispo, Leandro Santiago Padilla, Seidy Pedroso Santana, Natalie C. Parra, Jorge Roberto Toledo Alonso

**Affiliations:** 1Laboratorio de Biotecnología y Biofármacos, Departamento de Fisiopatología, Facultad de Ciencias Biológicas, Universidad de Concepción, Víctor Lamas 1290, Concepción P.O. Box 160-C, Chile; thramos@udec.cl (T.I.R.); cvillagui@outlook.es (C.A.V.-A.); spedroso@udec.cl (S.P.S.); natparra@udec.cl (N.C.P.); 2Grupo de Investigación en Sanidad Animal y Humana (GISAH), Carrera Ingeniería en Biotecnología, Departamento de Ciencias de la Vida y la Agricultura, Universidad de las Fuerzas Armadas—ESPE, Sangolquí 171103, Ecuador; 3School of Biological Sciences and Engineering, Yachay Tech University, Hda. San José s/n y Proyecto Yachay, Urcuquí 100119, Ecuador; nvispo@yachaytech.edu.ec; 4Faculty of Biological Sciences, Friedrich-Schiller-Universität, 07743 Jena, Germany; saintpad97@gmail.com

**Keywords:** interferons, IFN-α, IFN-β, IFN-γ, antiviral, antiproliferative, immunomodulator, PEGylation, formulation, encapsulate IFNs, drug delivery system, liposomes, polymeric micelles, microparticles, nanoparticles

## Abstract

Interferons (IFNs) are cytokines involved in the immune response that act on innate and adaptive immunity. These proteins are natural cell-signaling glycoproteins expressed in response to viral infections, tumors, and biological inducers and constitute the first line of defense of vertebrates against infectious agents. They have been marketed for more than 30 years with considerable impact on the global therapeutic protein market thanks to their diversity in terms of biological activities. They have been used as single agents or with combination treatment regimens, demonstrating promising clinical results, resulting in 22 different formulations approved by regulatory agencies. The 163 clinical trials with currently active IFNs reinforce their importance as therapeutics for human health. However, their application has presented difficulties due to the molecules’ size, sensitivity to degradation, and rapid elimination from the bloodstream. For some years now, work has been underway to obtain new drug delivery systems to provide adequate therapeutic concentrations for these cytokines, decrease their toxicity and prolong their half-life in the circulation. Although different research groups have presented various formulations that encapsulate IFNs, to date, there is no formulation approved for use in humans. The current review exhibits an updated summary of all encapsulation forms presented in the scientific literature for IFN-α, IFN-ß, and IFN-γ, from the year 1996 to the year 2021, considering parameters such as: encapsulating matrix, route of administration, target, advantages, and disadvantages of each formulation.

## 1. Introduction

Interferons are a family of cytokines whose functions have been known for more than six decades [1]. Their primary function is the stimulation of the immune system, triggering antiviral, antiproliferative, and immunomodulatory responses [2]. These proteins are critical effectors of innate and adaptive immunity, associated with activating a humoral and cellular response to different pathogens derived from neoplastic processes and other damage responses to the organism [3]. There are three main groups: type I, type II, and type III IFNs. Type I IFNs include eight different subtypes classified according to the Greek letters α, β, ε, ω, κ, δ, τ, and ζ [4]. Within type I, IFN-α and IFN-β stand out for their potent antiviral function [5], activated through a signaling cascade triggered by heterodimerization of IFN-α/β receptor 1 (IFNAR) of nucleated cells [6] (Figure 1). This pathway induces the expression of more than a thousand IFNs-stimulated genes (ISG[3]s) [7], whose generated proteins, such as 2–5 synthetase, protein kinase R (PKR), Mx protein, viperin, among others, play an essential role in the suppression of viral propagation [8]. IFN-α also possesses antiproliferative and immunomodulatory effects, through its function on apoptosis activation, mitotic cycle arrest, increased expression of major histocompatibility system (MHC) class I, stimulation of natural killer (NK) cells, and antigenic presentation [9]; as well as promotion of B and T lymphocyte differentiation [10,11].

IFN-γ is the only type II IFN. Type 1 innate lymphoid cells and NK secrete. This cytokine is in response to the recognition of infected cells [12]. Its primary function is regulating innate and adaptive immune responses, acting as a link between the two defense systems [13]. Additionally, it promotes antiviral immunity through its regulatory effects on the innate immune response [14]. The impact of IFN-γ as an antiviral on antigen-presenting cells (APCs) is to enhance the stimulation of the adaptive response to eliminate infection and generate protective memory for future infections [15]. This cytokine is a critical inducer of the Th1-type T cell response by optimizing the antigenic presentation process to MHC-I [16]. IFN-γ also increases MHC-II expression and the maturation of dendritic cells [17]. IFN-γ binding to its receptor (IFNGR) initiates a signaling cascade that activates its response [18] (Figure 2).

More recent scientific literature suggests that type I and II IFNs act synergistically by regulating the antiviral innate immune response and promoting the adaptive immune response while suppressing the detrimental functions of other immune cells and minimizing the collateral damage of infection [19]. The synergistic response of both IFNs initiates multiple waves of type I IFN production. At 12 h after infection, IFNβ is responsible for inducing inflammatory monocyte recruitment mediated by monocyte chemoattractant protein 1 (MCP-1), leading to IL-18-induced NK cell IFN-γ production [20]. At 48 h after infection, the second peak of type I IFN production (IFN-α and IFNβ) occurs: at the same time, there is an increase in IFN-γ released by NK cells, which is down-regulated by type I IFN [21]. This second peak of type I and II IFN likely acts in concert to promote antigen-presenting cell (APC) maturation, positive regulation of co-stimulatory molecules, antigen processing, and presentation towards a Th1 polarization, while at the same time suppressing innate lymphoid cell-mediated (ILC2) immunopathology [19].

In 2003, two independent research groups, Kotenko et al. and Sheppard et al., discovered type III IFNs through computational predictions [22,23]. They consist of four IFN-λ subtypes, which play a crucial role in mucosal antiviral defense [24]. The cytokine produces this immune response through a signaling pathway similar to type I IFNs but using an IFN-λ receptor 1 (IFNLR) [8] (Figure 3).

The diversity of biological activities that IFNs perform within the immune system has made their clinical use necessary since the 1980s [25], which is why different forms of these genetically engineered cytokines have been developed, allowing their application in various infectious, neoplastic, and autoimmune therapies [24]. There are two recombinant forms of IFN-α: 2a and 2b; two presentations of IFN-β: 1a and 1b; and one of IFN-γ: 1b [26].

The first IFN approved by the US Food and Drug Administration (FDA) for clinical use was IFN-α type I in 1986 [25]. Among the conditions treated with this molecule are AIDS-associated Kaposi’s sarcoma [27], hepatitis B [28], hepatitis C [29], condyloma acuminatum [30], herpes zoster [31], hairy cell leukemia [32], and a broad spectrum of ophthalmologic disorders. IFN-β was later approved in 1993 and is used to treat multiple sclerosis due to several overlapping mechanisms, such as decreasing the expression of major histocompatibility complex class II in antigen-presenting cells. It also induces interleukin-10 (IL-10) production, shifting the balance towards anti-inflammatory Th-2 helper T cells and inhibiting T-cell migration [33]; and reduces proinflammatory proteins’ production, such as IL-6 and TNF-α, which has made it a viable treatment for rheumatoid arthritis [34]. IFN-γ was approved in 1991 for therapy against chronic granulomatous disease and malignant osteopetrosis [35]. Regarding type III IFNs, there are currently no formulations approved to treat any disease, but they represent a potential candidate for clinical use because of their role in mucosal immunity [36].

IFNs’ therapy has encountered difficulties due to the size of the molecules, their sensitivity to degradation, and rapid elimination from the blood circulation [37]. The half-life of these cytokines is very short (2–3 h for IFN-α, 10 h for IFN-β, and 4.5 h for IFN-γ) [38,39,40]. This rapid clearance in blood makes administering high nonphysiological doses necessary, preferably parenterally [41]. This condition leads to substantial and unavoidable toxicity that limits its effectiveness, causes the occurrence of a variety of adverse events for the three types of IFNs [33,35,42], and weakens the quality of life of treated patients [43]. These limitations in clinical use have motivated the development of alternative delivery systems to achieve greater therapeutic efficacy and decrease its toxicity [24,35].

Research is now focusing on obtaining new drug delivery systems that aim to provide adequate therapeutic concentrations with lower toxicity for these cytokines, prolonging their half-life in the circulation and avoiding their degradation. The present review aims to compile the central encapsulation systems described in the scientific literature for type I and II IFNs, ranging chronologically in their development from PEGylation, liposomes, micelles to their most recent forms of encapsulation (microparticles and nanoparticles). The advantages and disadvantages of each encapsulation method used with these cytokines and the outlook for the most current emerging formulations. We will also exhibit the most important milestones that have emerged with these new formulations in their development towards clinical application, improving aqueous solubility, chemical stability, increasing pharmacological activity, and reducing side effects associated with the large doses required to achieve their pharmacological function.

## 2. IFN Delivery Systems

IFNs in their natural low molecular weight form (~20 kDa) are glycosylated proteins, but this post-translational modification does not play a functional role [44]. Obtaining IFNs was initially derived from leukocytes and lymphoblastoid lines, but extraction of proteins from natural producers suffers from limitations that limit regular large-scale production [45]. Recombinant DNA technology became an excellent option to produce these therapeutic proteins [46]. Recombinant IFNs are mostly non-glycosylated with identical biological activity to their natural counterparts [45]. They possess similar structures adopting a unique α-helix topology relative to other proteins [47]. These molecules possess amphipathic characteristics, with hydrophobic and hydrophilic regions conferring solubility [48]. Their instability, molecular size, hydrophilicity, low permeability, rapid clearance from circulation, and high susceptibility to degradation at low pH and in the presence of proteases have restricted their therapeutic application [49]. Formulations that protect IFNs from degradation and achieve prolonged releases with adequate biological activity are required [50,51]. Drug design systems that encapsulate therapeutic proteins maximize their biological potential, provide transport matrices that avoid the influence of weak non-covalent interactions (van der Waals forces) and electrostatic interactions that alter protein stability [52]. It also protects the cargo proteins from degradation by enzymes found at the administration site or during transport to the site of action, thus increasing their half-life [53].

New encapsulated formulations for IFNs have demonstrated several challenges, such as electrostatic interactions between IFNs (isoelectric point) and acidic end-groups of the encapsulation matrices (hydrolysis) with consequences on release; the pH of the formulation buffer and its variants with impact on solubility, stability, and aggregation [54]. Efficient encapsulation has been related to stabilizers that support particle size modulation and correlate with release patterns and biological activity [55]. Some of the encapsulations explored have shown the necessity to consider the polarity of the protein concerning the encapsulant. For example, in amphipathic nanovectors (polymeric micelles), molecules are encapsulated by stimulating protein-like polarity so that hydrophobic structures interact with hydrophobic parts of the system, and hydrophilic portions interact with hydrophilic regions [56]. Protein aggregation is related to hydrophobic interactions, encapsulating it through hydrophilicity to ensure stability (see Table 1) [57,58].

Some IFNs’ transport systems that have been studied and evaluated include PEGylation, self-assembled nanostructures such as liposomes and micellar systems, microparticles, and nanoparticles (metallic, polymeric, or hybrid) [59] (Figure 4). For developing these platforms, criteria of safety, biocompatibility, biodegradability, and compatibility of the encapsulating material with the drug must be considered and comply with the parameters that determine the functionality of a nanoparticle, such as its size, shape, and surface characteristics [60]. However, bringing this approach towards clinical application requires careful evaluation of efficacy, safety, and manufacturing [61].

## 3. PEGylation of IFNs

PEGylation was the first formulation aimed at improving the pharmacological properties of IFNs [62]. It consists of the covalent bonding of poly(ethylene glycol) (PEG) chains to a drug to increase its half-life, reduce its clearance, and improve its pharmacokinetics and pharmacodynamics [63]. PEG is FDA approved for systemic applications [63] due to its pharmaceutically relevant properties: increases IFN solubility and stability by decreasing proteolytic degradation; reduces renal clearance rate by increasing the size of the renal boundary molecule, decreases plasma clearance, improves the safety profile of the protein by protecting antigenic and immunogenic epitopes; and increases circulation time, high mobility solutions, and low toxicity [64,65].

Through covalent conjugation of PEG to IFN molecules, several formulations of PEGylated IFNs were developed using two types of conformation, linear and branched: Pegasys^®^ (Hoffmann La Roche Inc., Basel, Switzerland), conjugate IFN-α-2a to a 40 kDa branched-chain via amide linkages [66]; PegIntron^®^ and ViraferonPeg^®^ (Merck & Co., Inc., Whitehouse Station, NJ, USA), link IFN-α-2b to a 12 kDa linear molecule via a urethane linkage [67]; and Plegridy^®^ (Biogen, Cambridge, MA, USA), couple IFN-β-1a to a 40 kDa linear PEG chain with amide linkages [63]. PEG conjugation does not alter protein conformation significantly, but several aspects such as chain size and structure influence biological activity [63]. Increasing molecular weight increases the half-life, and coupling 30–40 kDa PEGs achieves this effect [67]. IFNs that bind to branched chains have a lower loss of biological activity than those conjugated to linear chains because binding to different amino acids binds a more significant amount of polymer [66,68,69]. The half-life and stability in branched PEGylations are higher because this structure decreases the glomerular filtration of proteins and protects their surface to a greater extent [70]. These forms of PEGylation with IFNs showed reduced excretion through the kidneys [71], with a five- to tenfold increase in half-life time, resulting in more stable drug concentrations in the plasma of patients [72], and replacing systemically applied IFNs [34,73]. This platform directly enhanced the drug’s pharmacokinetics, making possible less frequent dosing intervals of PEG-IFN-α on patients with chronic hepatitis B and C while still effectively reducing their viral load [29,74]. PEGylation of IFN-ß used in multiple sclerosis therapy resulted in a more comfortable regimen for the patient by reducing the dosage [74]. Similarly, PEG- IFN-γ conjugation has been evaluated, finding an increase in drug half-life of up to 32-fold in in vivo studies [75]. Currently, some formulations of PEGylated IFNs are in preclinical and clinical trial stages, including PEGylated IFNβ (preclinical testing completed) and PEGylated IFN-α (in preclinical studies), both from Bolder BioTechnology, as well as RogPEGinterferonα-2b (P1101) from PharmaEssentia (in preclinical studies) [45].

The loss of IFN activity caused by PEGylation is up to 80% of native IFN, which increases the amount of drug necessary to obtain an antiviral effect equivalent to that of native cytokine, and thus a more significant induction of toxicity [76]. Therapy with these encapsulated formulations can cause a range of adverse events, from mild to severe, such as diabetes, liver neoplasms, or psychotic disorders [34,77]. The decrease in bioactivity could not always increase the in vivo therapeutic efficacy of IFN [78], so treatments with these molecules are often unsatisfactory and should be discontinued [79].

Other forms of gene fusion conjugation have been evaluated as an alternative to PEGylation: PASylation and binding to human serum albumin (HSA) to increase the half-life of IFNs [80]. In the case of PASylation of IFN-α and IFN-β, a terminal polypeptide sequence rich in proline, alanine, and serine (PAS) was added [81], which increased stability, bioavailability, and biological activity, with minimal toxicity and immunogenicity [82,83,84] but did not reach the clinical phase [85]. Through the fusion of IFN-α-2b with HSA, the FDA-approved formulation Albuferon^®^ (Human Genome Sciences Inc., Rockville, Maryland, U.S. in collaboration with Novartis AG) was developed, which improved pharmacokinetics by increasing the half-life and maintaining its stabilization. This system had a prolonged serum half-life that allowed dosing at two- to four-week intervals. However, it was withdrawn from the market due to its toxicity [45].

In conclusion, although PEGylation of IFNs initially improved treatment efficacy, their toxicity has relegated the therapy to second-line status in most developed countries. Formulations still need to be developed using alternative strategies to increase the stability and reduce the clearance and toxicity of IFNs without compromising their biological activity [8].

## 4. Liposomes

Liposomes are spherical structures formed by one or more concentric lipid bilayers surrounding aqueous spaces [86]. They consist of phospholipids and cholesterol, formed by hydrophobic interactions and other intermolecular forces, and possess hydrophobic and hydrophilic regions [87]. Liposome-based drug delivery systems have shown unique characteristics to cross biological obstacles and improve pharmacodynamics [88]. Some of the advantages of this delivery system include biocompatibility, low immunogenicity, self-assembly ability, and the ability to transport drugs, such as IFNs, thereby reducing systemic toxicity and prolonging residence time in the circulation [86]. There are different liposomal formulations for encapsulating chemotherapeutic drugs, antifungals, and vaccines, currently approved by regulatory agencies for clinical application [89].

Gurari-Rotman and Lelkes reported the first encapsulation of IFN-α in multivesicular liposomes in 1982 [90]. Consequently, similar investigations were developed with IFN-γ [91,92]. Hume and Nayar demonstrated in 1989 that this encapsulation did not interfere with the molecule’s biological activity [93]. Some formulations were presented for type I [94,95] and type II IFN [96,97,98] from that year on. However, the encapsulation efficiency of these liposomes with IFNs variants did not exceed 50%, so their therapeutic use was not considered viable in 1998 despite the improved pharmacokinetics observed in murine models [99,100].

New formulations were developed at the beginning of the 21st century, using different strategies to improve encapsulation efficiency (see Table 1). Vyas and collaborators (2006) produced 20 µm multivesicular liposomes with IFN-α, achieving 75% encapsulation efficiency with the double emulsion method. In vitro studies confirmed that the system provided a sustained release of 6 days, with an abrupt initial release [101]. In the same year, Yang et al. achieved improved encapsulation efficiency of up to 80% by making 101 nm liposomes using the film hydration method. This formulation administered intramuscularly in *Kungming* mice allowed increasing the residence time of IFN-α at the injection site by up to 24 h. However, their result also showed a 10% reduction in the molecule’s activity and shorter sustained release time [102].

The most recent systems were synthesized by the film hydration method because of their higher encapsulation efficiency. Li and coworkers (2011) evaluated the pharmacokinetics of 172 nm liposomes with IFN-α in *Wistar* rats, finding higher bioavailability, maximum circulation time, and half-life time than systemic IFN-α [103]. In 2012, Li et al. encapsulated IFN-γ in liposomes with cyclic peptides. In vitro studies revealed selective liposome transport into hepatic stellate cells, and in vivo experiments in *Sprague–Dawley* rats evidenced increased half-life and antifibrinolytic activity of IFN-γ with decreased toxicity. Still, encapsulation efficiency was less than 35% [104]. In 2017 Jøraholmen et al. obtained liposomes with IFN-α conjugated to PEG molecules to increase their pharmacokinetics. Ex vivo studies in vaginal tissue of pregnant goats indicated high penetration of the formulation relative to the control. In this case, although PEGylation did not affect encapsulation efficiency, the release of the molecule resulted in virtually no release, with less than 1% of material released after 8 h [105]. In January 2021 Shamshiri MK, et al. presented a liposome design encapsulating IFN-γ targeted for an antitumor application [106]. In vitro and in vivo results indicated suitable attributes for Lip-F2 liposomal formulations (PEGylated liposomes) with tumor reduction and increased survival time in mice, but with cytotoxic effects in the C26 cancer cell line and colon carcinoma mouse models.

Despite the success that this form of encapsulation has had with different drugs, numerous challenges affect the effectiveness of liposomes in formulations with IFNs [88]. Hypersensitivity, opsonization, uptake by the Reticulo Endothelial System (RES), and immunosuppression are the primary negative responses of the immune system to liposomes. It is worth noticing that the production of lipid-based transport systems is expensive and that liposomes are not very stable because of their susceptibility to fusion, aggregation, or assembly without these cytokines [86]. For these reasons, there are currently no liposomal systems that encapsulate IFNs approved by regulatory agencies for clinical application [88].

## 5. Polymeric Micelles

Polymeric micelles are nanocarriers formed by the spontaneous arrangement of amphiphilic block copolymers in aqueous solutions [107]. Block copolymers are macromolecules of two or more different polymers joined by covalent bonds to form one structure. Its molecular conformation depends upon the number of blocks. Diblock copolymer consists of two homopolymers, while triblock copolymer has three homopolymers. More complicated architectures such as mixed arm block copolymers contain three polymer chains covalently joined at a common branching point [108]. Polymeric micelles possess a two-phase structure: a hydrophobic core and a hydrophilic corona that allows modifications to their surface [56]. Polymeric micelles have several advantages for drug delivery, such as their increased solubility, enhanced stability of the molecule, structural flexibility, capacity to encapsulate a wide range of therapeutics, and the possibility of adjusting their size at the nanometer scale [56,108,109]. Modifications in the corona make it possible to reduce their clearance by the RES, thus prolonging their circulation time [110]. In this way, it is feasible to decrease the drug dose and the toxicity associated with drugs such as IFNs [56].

There have not been many encapsulations attempts of IFNs with polymeric micelles. So far, the proposed formulations are recent and only use IFN-α (see Table 1). Liu and collaborators (2018) reported the first system for a systemic application in ovarian cancer, who encapsulated the protein in micelles of a self-assembled copolymer, consisting of poly (oligoethylene glycol polymethacrylate) (POEGMA) and N-(2-hydroxypropyl) methacrylamide (PHPMA). This formulation increased pharmacokinetics up to 83.8 h and showed antitumor activity without inducing toxicity in mice with ovarian tumors [57]. Wang and coworkers in 2020 created micelles of two elastin-like polypeptide building blocks using thermoregulated assembly. This approach increased the molecules’ half-life and showed antitumor activity in mice with ovarian tumors. However, the drug circulation at the systemic level was lower (54.7 h) [58].

These experiments did not evidence in their results the encapsulation efficiency of IFN-α; a parameter considered one of the primary challenges to overcome since the encapsulation percentage is usually low (between 30 and 50%) [56]. Particle stability was another problem, which they resolved by regulating the assembly with temperature changes. However, this process involves a genetic fusion of IFN-α with the polypeptide block copolymer sequence, so that the process includes extra steps compared to other encapsulation strategies [58]. The complexity of micellar systems, together with the lack of methods to validate drug release and integrity of formulations, has limited therapeutic approval by regulatory agencies [111].

## 6. Recent Encapsulation Forms of IFNs

Due to the lack of success of transport systems in bringing these compounds to a clinical stage, researchers have focused on using micro and nanoencapsulation to protect IFNs from degradation and extend their half-life. With this approach, small doses can be administered in a sustained manner with minimal side effects [45]. Microencapsulation and nanoencapsulation are a set of technologies that allow the trapping of active ingredients, also known as core materials, using a surrounding element (encapsulation or coating) [112]. The distinction between nanoencapsulation and microencapsulation is the size of the particles obtained, considering that nanoparticles have a size between 1 and 1000 nm, and microparticles have a diameter greater than 1000 nm [113]. Both technologies aim to create a physical barrier to protect the active ingredient from the external environment, allowing a controlled release over time. Therefore, avoiding damage due to high concentrations of the active ingredient in the system [114] while extending its pharmacokinetics and reducing the fluctuation of serum levels, maintaining convenient doses [115].

### 6.1. Microencapsulation

This technique makes it possible to protect sensitive drugs from the external environment in micrometer structures [114]. The products of this encapsulation are microparticles distinguishable as microspheres or microcapsules according to their internal constitution and morphology [116]. Microspheres act as reservoir systems that embed the active ingredient in the particle matrix. In contrast, microcapsules consist of matrix systems comprising the drug as the core and the particle material as the capsule shell [117]. Various subtypes of IFNs have used microsphere encapsulation for multiple purposes (see Table 1) [118].

#### 6.1.1. IFN-α

In 2002, Zhou and coworkers stabilized IFN-α in poly(lactide-poly(ethylene glycol) microparticles that showed sustained release and activation of antiviral activity for 11 days in vitro in vesicular stomatitis virus (VSV)-infected Wish cells [119]. Diwan and Park (2003) obtained poly (lactic-co-glycolic acid) (PLGA) microspheres that showed sustained release of PEG-IFN-α for up to 3 weeks in in vitro models. However, their results also showed high release peaks that could be related to the molecule’s toxicity. A demonstration in an in vivo model was lacking [120]. Sanchez et al. in 2003 studied the synthesis of PLGA particles, finding that microspheres’ encapsulation improved cytokine release kinetics, which extended for 96 days in an in vitro experiment [54].

The IFN-α micro formulation that came closest to being approved for the treatment of hepatitis C is the so-called Locteron, first reported by De Leede et al. (2008). It consisted of microspheres of the polyether-ether-ester copolymer Poly-Active encapsulating recombinant IFN-α-2b produced in the *Lemna* duckweed system [121]. Phase IIB clinical trials of this product showed that it had similar antiviral efficacy to PEG-IFN-α2b, with higher serum levels and decreased toxicity relative to the comparator [122]. However, injection site reactions and mild to moderate neutropenia did occur, comparable to those reported for PEG-IFN-α-2b and PEG-IFN-α-2a, respectively [73,123]. Due to its efficacy and considerable toxicity reduction, Biolex Therapeutics considered performing further phase III clinical trials; however, the firm filed for bankruptcy in 2012 [124]. No other company acquired the rights to Locteron [125]. There is no record of any phase III clinical trials in the US National Institutes of Health (NIH), clinicaltrials.gov, or FDA drug approvals for therapeutic use [126].

Beyond the relative success of Locteron, several investigations were continued in 2008 using other strategies. Thus, Sáez and coworkers encapsulated the protein in PLGA microparticles, with no changes in its physicochemical and biological characteristics during its release in vitro [127]. Zhang and coworkers fabricated PLGA microparticles, observing an increase in the residence time of IFN-α in serum up to 18 days and a sustained release of up to 12 days in studies in *rhesus* monkeys [128]. The alginate-chitosan microspheres of Zheng and coworkers also manifested a 4-fold increase in the half-life time of IFN-α. However, their bioavailability was reduced, with no improvement in peak blood concentration [129]. Zhou et al. proposed loading magnetite nanoparticles to PLGA microspheres for site-specific delivery. This modification also improved encapsulation efficiency compared to non-magnetic particles, but in vitro antiviral assays in Vero cells against VSV indicated a reduction in the molecule’s biological activity [130].

In 2010, Yang et al. obtained PLGA microspheres that exhibited a 7-day sustained release of IFN-α, maintaining its biological activity in the in vitro Wish/VSV cell system [131]. Li and coworkers (2011) increased the in vitro cumulative release for up to 23 days by encapsulating the cytokine in PLGA-PEG/polybutylene terephthalate (PBT) microspheres. In vivo assays in *Sprague–Dawley* rats of this formulation showed that plasma levels of the molecule were stable for 13 days, starting with a rapid release on the first day [132]. In 2014, Gulia et al. impregnated protamine sulfate to gelatin microspheres to increase the release time of IFN-α to 336 h and prolong the in vitro cytotoxic effect on ovarian cancer SK-OV-3 cells [133]. Chen et al. (2016) proposed a different structure by formulating chondroitin sulfate and PVP microneedles that had stability for two months and did not cause skin damage after subcutaneous administration in *SD* rats [134].

#### 6.1.2. IFN-β

Kamei and coworkers (2009) performed the first microencapsulation of this subtype in poly (methacrylic acid-ethylene glycol) particles for intestinal administration and demonstrated improved intestinal adsorption, release, and pharmacokinetics of this formulation in *Sprague–Dawley* rats [135]. In 2013, Kondiah et al. encapsulated IFN-β in trimethyl chitosan-poly (ethylene glycol)-methacrylic acid trimethyl methacrylate microparticles as a pH-sensitive transport system and administered orally. In vitro experiments indicated that 74% release of the cytokine at an intestinal pH of 6.8, and *New Zealand White* rabbits trials showed that the release profile exceeded 24 h [136]. This is the latest formulation reported in the literature for this protein.

#### 6.1.3. IFN-γ

IFN-γ was the first microencapsulated subtype, but the least successful so far, as research ceased before 2000. Cleland and Jones (1996) first encapsulated this molecule in PLGA microspheres, using trehalose to prevent denaturation under encapsulation conditions, maintaining the native conformation and in vitro biological activity [137]. In 1997, Conway and Alpar produced polylactide microspheres that exhibited sustained in vitro release for 400 h [138]. Building on this study, Eyles et al. (1997) evaluated the pharmacokinetics of the system when administered orally in *Wistar* rats, finding increased absorption compared to unencapsulated IFN-γ [139]. Yang and Cleland complemented in 1997 the two previous investigations by reporting that the cytokine was not adsorbed on PLGA and indicating that a high concentration of salts caused aggregation of the protein, preventing its correct release in vitro [140]. The problems reported by several authors regarding the protein’s stability for this encapsulation system led to the evaluation of alternative strategies such as the formulation in nanoparticles [137,138,140]. Further and more recent results for the encapsulation of this subtype are not yet available in the literature.

Despite the positive results obtained in in vitro and in vivo studies of microencapsulations, none of them were evaluated in clinical trials except for Locteron, since they presented drawbacks at different levels. The microencapsulation process achieved less than 60% encapsulation efficiency in several formulations [130,136], while a reduction in biological activity was observed in some cases [119,129,133]. In other formulations, the drug was released abruptly or incompletely [120,128,141]. For instance, a study conducted by Saez et al. in 2013 showed that IFN-α release in PLGA microparticles did not exceed 75% [142]. Most of the systems used the double emulsion/solvent evaporation method to obtain the microspheres. This process is challenging to scale up so that large-scale production of the formulations would be very costly and unstable [143].

### 6.2. Nanoencapsulation

Researchers worldwide have evaluated the possibility of encapsulating IFNs using nanoparticle systems since nanoformulations can improve its therapeutic index, especially in IFNs with a short half-life that therefore require frequent administration of high doses [50,51]. Nanoparticles are nanoscale structures that, like microparticles, can be capsules or spheres depending on their internal constitution [144]. These systems make it possible to simplify the administration of IFNs, improve their therapeutic effects and reduce their dose-related side effects without reducing their biological activity or changing the protein structure (see Table 1) [45].

#### 6.2.1. IFN-α

The first strategy used to transport IFN-α in nanoparticles was not an encapsulation but a drug conjugation to lysine-coated gold particles, developed by Aghdam and coworkers in 2008 [145]. Using this principle, Lee and coworkers (2012) coupled the antiviral to gold nanoparticles containing hyaluronic acid, which were selectively transported to the liver and exhibited similar biological activity to PEG-IFN-α-2b in a murine *BALB/c* mouse model [146].

These systems were unique; since years after the study by Aghdam and co-authors, research was mainly focused on encapsulations using polymeric nanoparticles. The main route of elimination of IFN-α is renal catabolism, with minimal amounts of IFN-α in the urine, while hepatic metabolism and biliary excretion are minor routes of elimination [147]. In 2011, Giri and coworkers obtained PLGA nanoparticles adsorbed with hepatitis B virus surface antigens. This configuration allowed the transport of the molecule into the liver without passing through the RES, maintaining a stable plasma concentration for 24 h. The distribution and opsonization of PLGA nanoparticles appeared to be influenced by size (174 nm) and surface characteristics with uniform distribution. When using Intravenous administration, the filtering effect of the pulmonary capillary bed removed large particles and aggregates. HBsAg-modified particles remained in the circulation longer than the uncoated counterpart, with a high blood concentration of the loaded drug for a prolonged period, with a targeting effect [148]. Feczkó et al. (2016) encapsulated PEG-IFN-α-2a in PEG-PLGA nanoparticles that released the cytokine in a sustained manner for four days in vitro [149]. In 2017, Cánepa et al. synthesized chitosan nanoparticles that exhibited comparable antiviral activity to commercial IFN-α in in vitro models. In the same study, preliminary in vivo assays in a *CF-1* mouse murine model showed that the molecule was detectable in the blood after one hour of oral administration of the formulation [150]. Imperiale et al. complemented this study in 2019 by demonstrating that the system provided bioavailability comparable to intravenously administered IFN-α-2b in murine *BALB/c* mouse models [151]. Kristó et al. (2020) developed a nanoparticle with a core/shell structure, whose core was composed of IFN-α associated with human serum albumin, and the shell was constituted by three consecutive layers of polystyrene-chitosan-polystyrene sulfonate-sulfonate, respectively. This system exhibited sustained release for ten days without reducing cytokine biological activity in a higher organism (*Pannon* rabbits) [152].

#### 6.2.2. IFN-β

Fodor-Kardos et al. in 2020 reported the first IFN-ß nanoencapsulation procedure. This research obtained nanometric particles with high encapsulation efficiency (>95%) to treat multiple sclerosis. Although in vitro studies *and* in hepatocytes of *Wistar* rats administered with the formulation showed no evidence of toxicity, they observed mild side effects in the kidney, such as whiteness and pyelectasis [153]. Gonzalez et al. (2021) studied the effect of intranasal administration of chitosan/cyclodextrin nanoparticles loaded with IFN-ß for the treatment of multiple sclerosis, finding that the nanoformulation was more effective than systemic administration of the cytokine in a murine model of *C57BL/6* mice with better availability and immunomodulatory effects [154].

#### 6.2.3. IFN-γ

Segura et al. in 2007 evaluated the macrophage activation capacity of IFN-γ encapsulated in human serum albumin nanoparticles, observing an increase in the bactericidal effect against *Brucella abortus* of macrophages activated by the formulation in *BALB/c* mice [155]. Yin et al. (2018) developed a nanoparticle with a core–shell structure encapsulating IFN-γ and doxorubicin for anti-melanoma therapy. In vivo studies in *C57BL/6* mice determined a half-life extension of the cytokine up to 48 h without producing toxicity in vital organs at the doses administered [156].

Nanoencapsulation exhibits better functionality than microencapsulations showing increased drug protection, more excellent stability, superior loading capacity, encapsulation efficiency, sustained release profile, and improved bioavailability of the active ingredient [157,158]. Research and development in the nanotechnology field have significantly advanced over the past few years. Still, the selection of nanocarrier systems and their potential applications can confuse researchers without prior knowledge in the field [159]. The significant challenges in the formulation of drug delivery systems are to control drug release and avoid the opsonization of the particle. Therefore, it is necessary to analyze the different drug delivery mechanisms and methods to increase their bioavailability. Biological barriers and their impact on the transport of the active ingredient must also be taken into account [159]. On the other hand, demonstrating its efficiency limits the implementation of a sustained release and transport system. At the same time, it must have minimal levels of cytotoxicity and immunogenicity [160].

**Table 1 pharmaceutics-13-01533-t001:** Comparative table summarizing all forms of IFN-α, IFN-β, and IFN-γ delivery systems described in the scientific literature from 1996 to March 2021.

IFN Type	Encapsulating Matrix	Route of Administration	Encapsulation Method	Physical Properties	Formulation Objective	Advantages/Disadvantages	Ref.
**IFN-α**	Microspheres of LEAVE	In vitro	Double emulsion/solvent evaporation	Size = 186 µm	Stabilization of IFN-α on PELA particles with sustained release and retention of antiviral activity for up to 11 days in in vitro studies.	A: stabilization of IFN in the matrix D: initial burst release	[119]
	PLGA microspheres	In vitro	Double emulsion/solvent evaporation	Size = 1.8 µm	Sustained in vitro release of methoxy-PEG-IFN-α for up to 3 weeks, although they exhibited high release peaks.	A: solubility maintained D: initial burst release	[120]
	PLGA/ poloxamer	In vitro	Oil-in-oil solvent extraction	Size = 40 µm	Evaluation of microparticles and nanoparticles as an in vitro controlled release system. The MPs released IFN for up to 96 days.	A: integrity and activity of the molecule D: initial burst release	[54]
	Multivesicular liposomes	In vitro	Double emulsion/solvent evaporation	Size = ~20 µm	Development of a system for controlled and sustained release of PEG-IFN-α for up to 6 days in vitro.	A: high stability and encapsulation efficiency D: initial burst release	[101]
	Uni- and multivesicular liposomes	Intramuscular	Film hydration-dilution	Size = 101 nm	Prolonged retention of IFN-α-2b for up to 24 h at the application site after intramuscular administration in *Kungming* mice.	A: high retention at the application site D: loss of activity	[102]
	Lysine-coated gold nanoparticles	In vitro	Chloroauric acid and borohydride reduction	Size without IFN = 10 nm	in vitro transport of IFN-α on gold nanoparticles coupled to lysine found on the particle surface.	A: stable conjugation in water D: modification of the carboxyl groups of the molecule	[145]
	Poly(ether-ester) microspheres (Poly-Active)	Subcutaneous	Double emulsion/solvent evaporation	Size = ~30 µm	Phase IIB clinical study of Locteron^®^, a 14-day dose–response sustained-release formulation, well tolerated by patients at a dose of 80 µg.	A: significant decrease in adverse events D: scarce report of its physicochemical characterization	[121]
	PLGA microspheres	In vitro	Double emulsion/solvent evaporation	Size = 28.1 µm	Encapsulation of IFN-α in PLGA microparticles in vitro. No changes were detected in the physicochemical and biological characteristics of the molecule released by diffusion for 24 h at 37 °C.	A: uniform size distribution D: IFN instability	[127]
	PLGA microspheres	Intramuscular	Double emulsion/solvent evaporation	Size = 81.23 µm	Increased residence time of IFN-α in serum up to 18 days, and sustained release with activity up to 12 days in studies in *rhesus* monkeys.	A: increase in circulation time in vivo D: loss of biological activity	[128]
	Alginate microspheres chitosan	Intramuscular	Coacervation	Size = 2.18 µm	Evaluation of pharmacokinetics in *ICR* mice, revealing a 4-fold increase in the half-life of IFN-α, with no increased peak concentration, and reduced bioavailability	A: increase in maximum serum concentration D: low encapsulation efficiency	[129]
	PLA and PLGA microspheres	In vitro	Double emulsion/solvent evaporation with magnetite nanoparticles inclusion	Average size = 2.5 µm Size distribution = 0.5–3.5 µm	Particle loading with magnetite for site-specific delivery. In vitro antiviral assays in Vero cells against vesicular stomatitis virus indicated a slight reduction in the antiviral activity of the particles.	A: particle direction using magnetic field D: low encapsulation efficiency	[130]
	PLGA microspheres	In vitro	Double emulsion/solvent evaporation	Size distribution = 40.54–115.62 µm	Sustained release maintains the molecule’s biological activity for up to 7 days in in vitro studies in *Wish* cells against vesicular stomatitis virus.	A: high encapsulation efficiency D: in vivo performance was not evaluated.	[131]
**IFN-α**	PLGA-PEGT/PBT microspheres	Subcutaneous	Double emulsion/solvent evaporation	Size = 28.94 µm	Extended cumulative release for up to 23 days in vitro, conforming to zero-order kinetics. Plasma levels were stable for 13 days in *Sprague–Dawley* rats, starting with a rapid release on day 1.	A: high encapsulation efficiency D: initial burst release	[132]
	PLGA nanoparticles with adsorbed HBV antigens	Intravenous	Double emulsion	Size = 174 nm PZ = +30 mV	System aimed at treating hepatitis B. Studies in *BALB/c* mice indicated that nanoparticles transport IFN to hepatocytes, with good systemic circulation.	A: site-specific transport D: low encapsulation efficiency	[148]
	Liposomes	Intramuscular	Film hydration	Size = 82–172 nm PDI < 0.35	Increased half-life, peak time, and bioavailability of encapsulated IFN-α-2b in *Wistar* rats.	A: accumulation in the liver D: non-uniform size	[103]
	Gold nanoparticles plus hyaluronic acid (HA)	Intravenous	Chloroauric acid reduction with citrate and reductive amination of HA	Size = 52.23 nm PDI = 0.089	Selective transport to the liver for HCV treatment. Biological activity of IFN-α is similar to PEG-Intron in vitro (Daudi), in vivo (*BALB/c* mice).	A: serum stability D: slow initial release	[146]
	Protamine sulfate-impregnated gelatin microspheres	In vitro	Emulsion polymerization with glutaraldehyde as a crosslinker	Size = 28.94 µm	Protamine sulfate impregnation to increase the release time of IFN-α to 336 h and prolong the cytotoxic effect in vitro in ovarian cancer *Skov3* cells	A: almost complete release D: no correlation with cytotoxicity	[133]
	Chondroitin sulfate and PVP	Intradermal	Two-solution system in polydimethylsiloxane molds	Arrangements of 12 × 12 microneedles. Dimensions: 680 × 380 μm	Transport of IFN-α in microneedles. In vivo studies (*SD* rats), the needles have good stability for two months and do not cause skin damage.	A: no injections required D: limited stability over time	[134]
	PLGA and PEG-PLGA nanoparticles	In vitro	Double emulsion/solvent evaporation	Size = 104–129 nm	Evaluation of sustained release of IFN-α under in vitro conditions: phosphate-buffered saline and blood plasma.	A: sustained and stable release D: in vivo pharmacokinetics not evaluated.	[149]
	Chitosan nanoparticles	Evaluation of the oral route	Ionotropic gelation	Size = 36 nm PZ = +30 mV	Nanoparticles for oral administration, with in vitro antiviral activity (MDBK) comparable to commercial IFN-α. IFN levels in plasma 1h after in vivo inoculation (in *CF-1* mice).	A: high encapsulation efficiency D: non-specific release in the stomach	[150]
	PEGylated Liposomes	Franz Cell Diffusion System	Film hydration	Size = 181 nm PZ = −13 mV	Formulation for treatment of human papillomavirus. No in vitro release. Ex vivo studies in goat vaginal tissue with high penetration of the molecule into the tissue.	A: crosses mucosa D: in vitro and ex vivo release was not correlated	[105]
	POEGMA-PHPMA copolymer micelles	Intravenous	Self-assembly of copolymer blocks	Size = 64.9 nm	Formation of micelles by self-assembled copolymer blocks that encapsulated IFN-α, with increased half-life up to 83.8 h, and antitumor activity in mice with ovarian tumors	A: effective tumor suppression D: decrease in biological activity	[57]
	Chitosan nanoparticles	Oral	Ionotropic gelation	Size = 36 nm PDI = 0.47 Potential Z = +30 mV	Evaluation of oral administration of nanoparticles. In vitro (Caco-2:HT29-MTX (9:1)) and in vivo (*BALB/c* mice) studies confirmed improved pharmacokinetics and bioavailability.	A: crosses intestinal epithelium D: no analysis in disease models	[151]
	Core-shell nanoparticles; core: HSA-IFN-α, shell: PSS-CS-PSS	Subcutaneous	Core: aqueous precipitation; shell: layer-by-layer assembly	Size = 100 nm PZ = −50 mV	Sustained-release after ten days in *Pannon* rabbits, with biological activity similar to lyophilized HSA-IFN-α.	A: bioactivity maintained D: PSS is not biocompatible	[152]
	Elastin-like copolypeptide micelles	Intravenous	Self-assembly of two copolypeptide building blocks	Size = 48 nm	Formation of micelles by blocks of two self-assembled polypeptides that encapsulated IFN-α, with an increase in its half-life up to 54.7 h, and antitumor activity in mice with ovarian tumors.	A: efficient accumulation in tumors D: encapsulation efficiency is not reported.	[58]
**IFN-β**	Poly(methacrylic acid-ethylene glycol) microparticles	Direct intestinal	UV polymerization using TEGDMA as crosslinker	Size < 53 µm	Encapsulation for intestinal delivery of IFN-ß. In vitro and in vivo results in *Sprague–Dawley* rats showed sustained release and improved pharmacokinetics.	A: pH-sensitive behavior D: incomplete release	[135]
	TMC-PEGDMA-MAA microparticles	Oral	Suspension polymerization by free radicals	Size = 1–3.5 µm at intestinal pH (6.8)	pH-sensitive oral transport system for the treatment of multiple sclerosis. Most of the IFN-ß was released in vitro at intestinal pH. Release profile in *New Zealand White* rabbits exceeded 24 h.	A: pH-sensitive D: in vitro and in vivo release was not correlated	[136]
	PLGA and PEG-PLGA nanoparticles	Subcutaneous	Double emulsion/solvent evaporation	Size = 145 nm and 163 nm PZ = 17.7 and 18.8 mV	Treatment of Multiple Sclerosis. No toxicity in vitro, in vivo studies in *Wistar* rats showed mild toxic effects such as pale kidney and pyelectasis.	A: high encapsulation efficiency D: mild toxicity	[153]
	Chitosan nanoparticles/cyclodextrin	Intranasal	Gelation	Size = 206 nm PZ = 20 mV PDI = 0.13	Nasal administration of the formulation for treating multiple sclerosis, with greater effectiveness, than free IFN-β in *C57BL/6* mice with sclerosis.	A: reduction in encephalomyelitis D: no CD4+ lymphocyte downregulation	[154]
**IFN-γ**	PLGA microspheres	In vitro	Double emulsion/solvent evaporation	Size = 30–50 µm	Stabilization of IFN-γ in microparticles, maintaining the native conformation and biological activity of the protein.	A: bioactivity maintained D: encapsulation destabilizes the protein	[137]
	PLA microspheres	Oral	Double emulsion/solvent evaporation	Size = 1.27 µm	Sustained release in vitro for 400 h and increased absorption when administered orally in *Wistar* rats.	A: increase in porosity D: delayed release	[138]
	Liposomes	Inhalation	Freezing, thawing	Size = 170–180 nm	It demonstrated that encapsulation of IFN-γ and liposomal muramyl tripeptide with chitosan activated alveolar macrophages and increased survival in the treated group. In vivo study in a murine model.	A: increase in the activation of alveolar macrophages. D: loss of biological activity	[97]
	BSA nanoparticles	Intraperitoneal	Coacervation and chemical crosslinking	Size = ~340 nm PZ = −19.6 mV	Evaluation of macrophage activation for *Brucella abortus.* It increased the bactericidal effect of IFN-γ-activated macrophages in vitro and in vivo (*BALB/c* mice).	A: increased biological activity D: extended-release only for 20 h	[155]
	Liposomes with cyclic peptides	Intravenous	Film hydration	Size = 83.5 nm PDI = 0.067	Selective liposome transport to hepatic stellate cells increased half-life and antifibrotic activity of IFN-γ with fewer adverse effects in *Sprague–Dawley* rats.	A: selective transport to hepatic cells D: low encapsulation efficiency	[104]
	PLGA core–shell nanoparticles containing IFN-γ and doxorubicin.	Intravenous	Nanoprecipitation	Size = ~100 nm	Melanoma immunotherapy. Female *C57BL/6* murine model, free IFN at 8 h, encapsulated cleared after 48 h inoculated in mice. There was no toxicity in vital organs.	A: temperature-sensitive behavior D: conditional encapsulation efficiency	[156]
	PEGylated Liposomes	Intravenous	Thin-film hydration and extrusion	Size = 135 nm PDI = 0.05	Preparation of IFN-γ-containing liposomes for colon cancer treatment. Sustained release in vitro for 144 h with an abrupt onset and increased cytokine-activated antitumor immune response in *BALB/c* mice with *C-26* tumor cells.	A: significant induction of the antitumor response D: low encapsulation efficiency	[106]

The current review showcases an updated summary of all forms of encapsulation including liposomes, micelles, microparticles and nanoparticles for IFN-α, IFN-β, and IFN-γ cytokines from 1996 to March 2021, considering the following parameters: encapsulating matrix, route of administration, encapsulation method, physical properties, target, advantages, and disadvantages of each formulation. Table 1 summarizes these systems.

## 7. Discussion

Although different research groups have presented various formulations to encapsulate IFNs, to date, there is no formulation approved for use in humans. Nevertheless, the importance of encapsulating IFNs is evident, considering new routes of administration. Since their first approval for clinical use in 1986, IFN formulations have improved since they are clinically valuable drugs [161]. IFNs are crucial elements in human humoral and cellular defense mechanisms and have shown clinical effectiveness against viral infections, various cancers, and neurodegenerative diseases by limiting virus replication, reducing tumor masses, controlling symptoms of autoimmune conditions, and prolonging survival [162]. They have been used as single agents or in combination treatment regimens, demonstrating promising clinical results, resulting in 22 different formulations approved by regulatory agencies. Three have been withdrawn from the market [163]. The 163 clinical trials currently active with IFNs reinforce their importance as therapeutics for human health [164]. On the economic side, total market sales of IFNs reached $6.9 billion in 2019, and these figures are expected to grow in the future due to the increasing incidence of viral and chronic diseases, and the growing adoption of biosimilars for potential therapeutics or prophylaxis of future pandemics, among others [45].

IFNs are broad-spectrum antivirals that are effective against viruses of recent interest, such as MERS-CoV [165], SARS-CoV [166], and SARS-CoV-2 [167]. These proteins exert autocrine or paracrine action on surrounding cells [45]. In an uninfected cell, binding of IFN to its receptor and subsequent IFN signaling renders the cell refractory to viral infection. In an infected cell, this signaling can suppress viral replication and decrease the release of viral progeny from the cell [168]. During initial infection in tissue, paracrine signaling can prevent the spread of infection by reducing the number of susceptible cells near the site of infection [169]. Therefore, a sustained release of IFN into the tissue, achieved through a transport system such as nanoencapsulation, is crucial in treating viral infections [170].

Immune system activation against oncogenesis and the control of tumor development by IFNs has enabled their use in treating neoplastic diseases. These antitumor effects are due to their ability to inhibit cancer cell growth by triggering apoptosis and cell cycle arrest [171] (Figure 5 and Figure 6). Since the proteins are administered in high doses to prevent rapid clearance, data predict that the antiproliferative and apoptotic activity would be more significant. However, research also shows that antitumor efficacy does not increase with an increasing amount of IFN administered, so small doses with minimal adverse events are more beneficial than higher doses [172]. Achieving optimal therapeutic response requires sustained transport and release systems due to the cytokine’s short half-life [8]. In this regard, nanoencapsulation allows for improved pharmacological activity without increasing the doses administered [173].

IFNs have also been used to treat various autoimmune disorders because of their ability to modulate innate and adaptive responses, both humoral and cellular [9]. Research showed that nanomedicines could actively and efficiently cross the blood–brain barrier (BBB) and penetrate deeply into diseased brain tissues (Figure 7). In addition, there is an association between the use of nanoformulations with enhanced resistance, stability, surface area, and sensitivity [174,175]. The delivery of nanoencapsulated IFNs into the central nervous system (CNS) has significantly improved current systemic immunomodulator-based therapies for autoimmune diseases. Intranasal administration of the nanoformulation resulted in a significant reduction in the cytokine’s effective dose [154].

IFN administration has been mainly intravenous and intramuscular. Researchers expected that elevated serum interferon titers would correlate with its therapeutic efficacy and that interferon levels in the interstitial fluid of target or effector cells would provide more information. Still, this connection does not exist [45]. Moreover, because the kidneys rapidly filter them, elevated serum titers are accompanied by considerable renal clearance. Thus, several factors converge, demonstrating a modest therapeutic efficacy of interferon observed so far, such as the inadequacy of routes of administration and its rapid systemic elimination.

Currently, these recombinant proteins, capable of activating mucosal immunity, are presented as ideal candidates in first-line treatments for acute viral infections, respiratory tract or sexually transmitted infections, and autoimmune conditions [176]. However, each specific physiological target exhibits some particularities requiring special considerations before their design.

We believe that one of the most appealing and novel routes for encapsulated interferon formulations could be intranasal, directly impacting the three biological actions of these cytokines (antiviral, antiproliferative, and immunomodulatory). The nasal mucosa provides the first-line defense against inhaled pathogens; it is the epithelial barrier to most infectious agents, especially respiratory viruses [177]. It presents lymphoid tissue, which contains abundant immune cells, such as B cells, CD4+ and CD8+ lymphocytes, and dendritic cells [178]. Its pseudostratified ciliated columnar epithelium lines most of the respiratory mucosa, nasal passages, nasopharynx, bronchial tree and performs physical, chemical, and immunological barrier functions [179]. Lymphoid cells and organs cover their surface is covered providing protective antibodies and cell-mediated immunity [180]. Mucosal epithelial lineages include epithelial cells that produce antimicrobial proteins (defensins and lectins); and goblet and columnar epithelial cells that release mucins [181]. These epithelial cells can also function as specialized antigen-presenting cells [182]. Viral infection induces the epithelium host defenses. Plasmacytoid dendritic cells (pDCs) detect viruses through Toll-like receptors, and pattern recognition receptors (PRR) activation triggers the production and release of type I and III interferons and other proinflammatory mediators, which initiate the host innate and adaptive immune response [183].

These mechanisms make the nasal mucosa a very attractive route of administration for inhalation formulations as they represent the preferred site for invasion of respiratory viruses or neoplastic entities at the mucosal level or near the CNS [184]. The nasal mucosa can be used for noninvasive drug delivery and constitutes a tissue well supplied by blood vessels [185]. It ensures rapid absorption of most drugs, can generate high systemic levels, and avoids the first-step (hepatic) metabolism characteristic in oral administration [186]. For this reason, applications of interferons in inhaled formulations have been used [187]. These therapeutic variants demonstrate a reduction in viral load, symptom relief and shorten the duration of viral respiratory infections such as bronchiolitis, pneumonia, and acute upper respiratory tract infections [188]. Since 1973, the British Medical Research Centre confirmed that IFNs could prevent and treat respiratory virus infections by activating the nasal mucosa using the inhalation route [189]. In 2003, during the SARS outbreak, an animal study (rhesus monkey) revealed that recombinant human IFN-α2b aerosol could prevent SARS-CoV infection by inhibiting virus infection and replication [190]. Other studies showed that recombinant human IFN-α2b aerosol reduced the infection rate of the respiratory syncytial virus, influenza virus, adenovirus, and SARS-CoV [191]. In turn, IFN-γ has been administered in diseases such as cancer, tuberculosis, hepatitis, chronic granulocytic disease, osteopetrosis, scleroderma, atypical mycobacteria, among others [188,190,192]. Its aerosolized use has been proposed as a methodology for organ-specific release cytokine therapy in infected airways [193,194], so both proteins’ safety profile and efficacy are widely known.

The most attractive formulations considering the activation pathway of interferons with potential for the intranasal route would be micro and nanoparticles. Still, as we already presented in the section on microparticles, these show evidence that they affect the integrity of their active ingredient, so the best encapsulation option for these proteins would be nanoparticles. These formulations at the nanometric level overcome the physical barrier of the mucous membranes and achieve a prolonged retention time on the cell surface, penetrating effectively and accumulating on the epithelial surface; they also protect the active principle from biological and chemical degradation [159]. Combining nanoparticles with absorption enhancers, functional excipients that improve penetrability at biological barriers, and enhancers that temporarily open tight intercellular junctions has been suggested, used in various nasal nanoformulations [195].

Currently, our research group is working on obtaining new drug delivery systems, which aim to provide adequate therapeutic concentrations for type I and II interferons, prolonging their half-life time in the circulation. Within these designs, nanoformulations are novel elements as drug carrier systems that allow: a sustained local release, a controllable long-duration administration, an interaction between different proteins that preserve the structural stability of the drug and its biological activity, which are novel features to be used with interferons.

## 8. Conclusions

There is great interest in achieving the encapsulation of interferons due to the diversity and effectiveness of their biological functions and the wide range of applications on the three major groups of immune system conditions, infectious, proliferative, and autoimmune. Researchers have applied various delivery methods, categorized as particle delivery systems, including micro and nanoparticles, liposomes, mini pellets, cell carriers, PEGylated IFNs, etc.

Nanoparticulate systems are very successful as a tool for developing peptide and protein delivery, capable of enhancing the efficacy of established drugs and new molecules. Due to their sustained release properties, subcellular size, and biocompatibility with tissues, these formulations have shown promise for the encapsulation of IFNs, allowing the bioavailability of drugs and improving the pharmacokinetic profile of other drugs for biomedical purposes. To date, an incredible amount of research has demonstrated the usefulness of nanoparticles in the formulation of new drugs and the protection they confer on mucous membranes and biological fluids by favoring penetration into cells. The final success in finding a nanoencapsulated formulation for interferons will be to prove their therapeutic potential and demonstrate their safety by integrating the research results with the pharmaceutical industry. We also know that selecting an appropriate route of administration will have a marked influence on the outcome of the proposed formulation, and we believe intranasal drug transfer could contribute to this outcome.

This review not only provides an up-to-date summary of all the forms of encapsulation that exist in the literature for interferons but is also a helpful starting point for new projections of the formulation of these cytokines and their contribution to their successful clinical application.

## Figures and Tables

**Figure 1 pharmaceutics-13-01533-f001:**
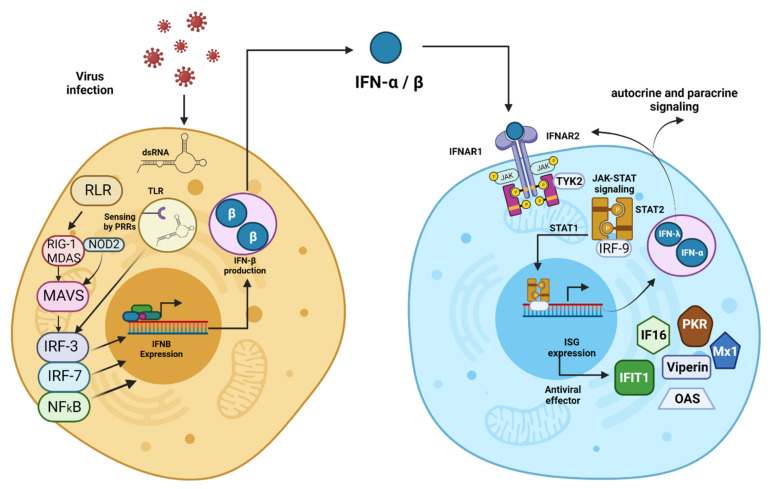
Type I Interferons Induction and Functions. Type I IFNs are first induced intrinsically in infected cells through a process of host cell recognition of segments of DNA or RNA or other viral macromolecules called pathogen-associated molecular patterns (PAMPs). PAMPs recognition as non-self-starts with their binding to specific cellular pathogen recognition receptors (PRRs), such as Toll-like receptors (TLRs), RIG-I-like receptors (RLRs), NOD-like receptors. Receptor-ligand binding triggers the type I IFN induction cascade via NFκß, resulting in the activation and translocation to the nucleus of IFN regulatory factors IRF3 and IRF7, which induce the expression of type I IFNs. The expressed cytokine is exported to the extracellular milieu and binds to IFNAR, a heterodimeric receptor consisting of two subunits, IFNAR1 and IFNAR2. The molecule forms a trimeric structure with the receptor, thus activating the proteins Janus kinase 1 (JAK1) and tyrosine kinase 2 (TYK2). These proteins activate the signal transducers and activators of transcription 1 and 2 (STAT1 and STAT2), which induce the transcription of interferon-stimulated genes (ISGs) by forming a complex with IRF9. These ISGs encode antiviral effectors (PKR, Mx1, OAS, etc.) and activate type I IFN production, thus triggering autocrine and paracrine signaling. Cells from the innate immune system, such as dendritic cells (DCs) and macrophages, produce type I IFN after sensing pathogen components using various PRRs found on the plasma membrane, in endosomes, and throughout the cytosol. Created with BioRender.com.

**Figure 2 pharmaceutics-13-01533-f002:**
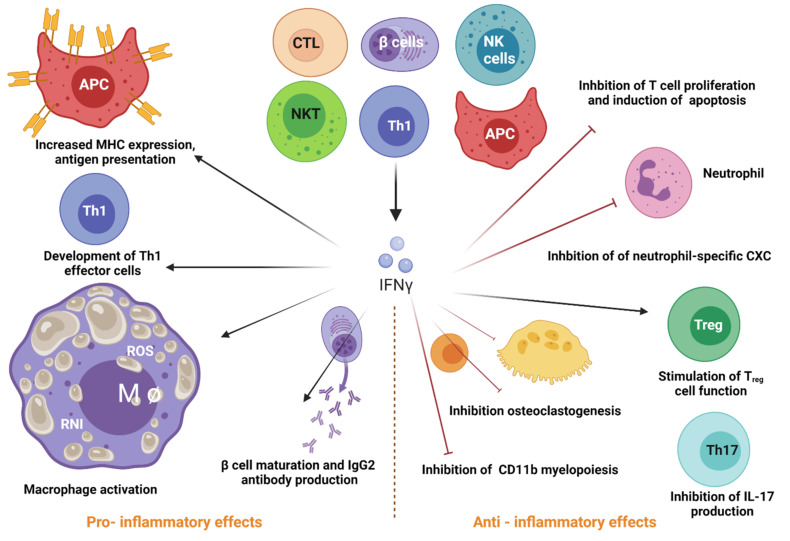
Roles of Type II Interferons. IFN type II is a pleiotropic cytokine that participates in viral response and regulating innate and adaptive immune responses. NK cells, T cells, B cells, APC release this cytokine to function both as an inducer (pro-inflammatory) and a regulator (anti-inflammatory) of immune responses. Regarding the pro-inflammatory effects, IFN-γ has a strong macrophage-activating activity and induces B cell maturation and IgG2 production. Additionally, this cytokine stimulates antigen presentation via MHC, development of Th1 effector cells, and cell function of T_reg_ cells. The anti-inflammatory effects of IFN-γ include the inhibition of T cell-dependent osteoclastogenesis and production of IL-17, which leads to decreased levels of neutrophil-specific CXC chemokines and limited mobilization of neutrophils. Furthermore, type II IFN inhibits myelopoiesis of CD11b^+^ leukocytes, T cell proliferation and induces apoptosis by secreting nitric oxide (NO) and indoleamine 2,3-dioxygenase (IDO). All these anti-inflammatory properties contribute to the protective role of IFN-γ against autoimmune diseases. Created with BioRender.com.

**Figure 3 pharmaceutics-13-01533-f003:**
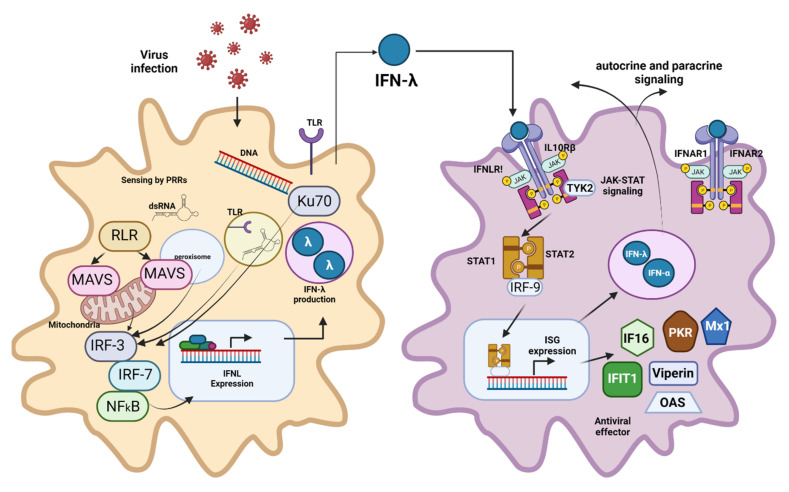
Type III Interferons Production and Activity. Type III IFNs act on epithelial cells and tissue-resident neutrophils, dendritic cells, macrophages, B cells, and plasmacytoid dendritic cells. Like the type I IFN pathway, IFN-λ induction starts with PRR (TLR and RLR) recognition of the respective PAMPs. Receptor-ligand binding triggers the IFN-λ induction cascade via NFκß, resulting in the activation and translocation to the nucleus of IFN regulatory factors IRF3 and IRF7, which induce the expression of type III IFN. After its release to the extracellular milieu, IFN-λ binds to its heterodimeric receptor (IFNLR), which consists of two subunits: α-subunit (IL28RA) and β-subunit (IL10RB). IFN-λ-IFNLR trimeric complex formation leads to the activation of JAK1 and TYK2, followed by the phosphorylation of STAT1 and STAT-2. Afterward, STAT1 and STAT-2 translocate to the nucleus and induce the expression of hundreds of ISGs with antiviral activity. Type I and III IFNs both show a complex mechanism of feedback loops, leading to autocrine and paracrine signaling. Even though epithelial cells are the primary source of type III IFN, macrophages, monocytes, and dendritic cells can also secrete them. Created with BioRender.com.

**Figure 4 pharmaceutics-13-01533-f004:**
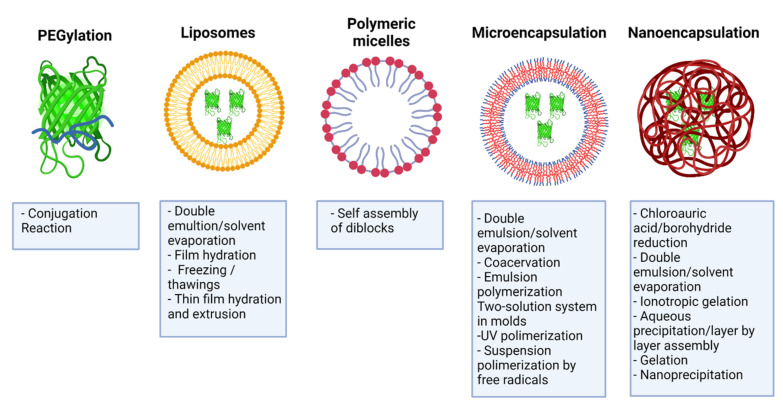
Encapsulation methods and IFN-delivery system. Summary of the different transport systems for type I and II IFNs, including PEGylation, liposomes, micellar systems, self-assembled nanostructures, microparticles, and nanoparticles. Created with BioRender.com.

**Figure 5 pharmaceutics-13-01533-f005:**
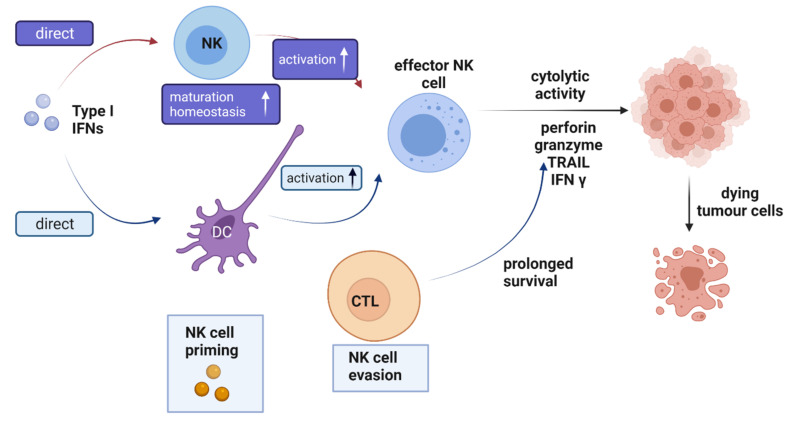
Activation pathways are regulated by type I interferons (IFNs) in the antitumor response. The antiproliferative activity of type I IFNs has anti-angiogenic effects on tumor vascularization, increased cytotoxicity, and survival of NK cells. These cytokines induce the generation and survival of cytotoxic T lymphocytes, memory CD8 T, and maturation of dendritic cells. Type I IFNs influence the maturation, homeostasis, and activation of NK cells, eliminating tumor cells through other immune cells or cells of the tumor microenvironment. Dendritic cells play an essential role in recognizing and presenting the various antigens that trigger the activation cascade. Another indirect effect of type I IFNs on NK cells in a tumor environment is the modulation of surface molecules on CD8 + cytotoxic T lymphocytes (CTL) [NCR1 ligands; classical and non-classical major histocompatibility complex class I (MHC I)] with evasion of CTLs from NK cell-mediated killing. Created with BioRender.com.

**Figure 6 pharmaceutics-13-01533-f006:**
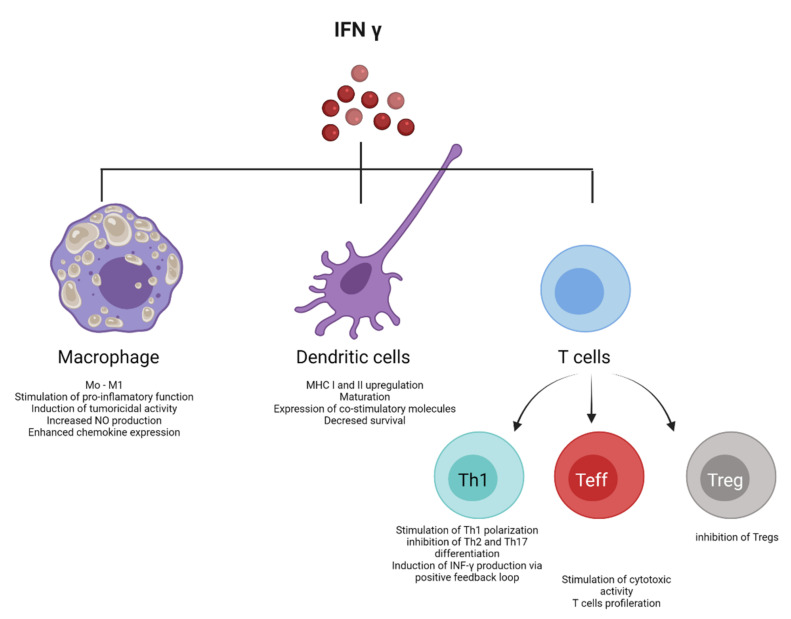
Role of IFN-γ in the antiproliferative response. Interferon-γ interacts with various cells in a tumor microenvironment to initiate the production of the cytokine itself. Some of these cells are T lymphocytes, macrophages, and dendritic cells. Macrophages: the protein stimulates the polarization of macrophages towards a proinflammatory phenotype by increasing the secretion of chemokines. Dendritic cells: increases the maturation of these cells, positive regulation of MHC I and II with increased IRF1 expression, and decreased IFN-γ-dependent dendritic cell survival. T cells: stimulates their differentiation with Th1 polarization. IFN-γ causes positive feedback, increasing their production in Th1 cells and inhibiting differentiation towards Th2 and Th17. Maturation of virgin T cells to effector CD8 + T cells requires IFN-γ. IFN-γ is the primary cytotoxic molecule secreted by these cells. IFN-γ inhibits immunosuppressive regulatory T cells. Created with BioRender.com.

**Figure 7 pharmaceutics-13-01533-f007:**
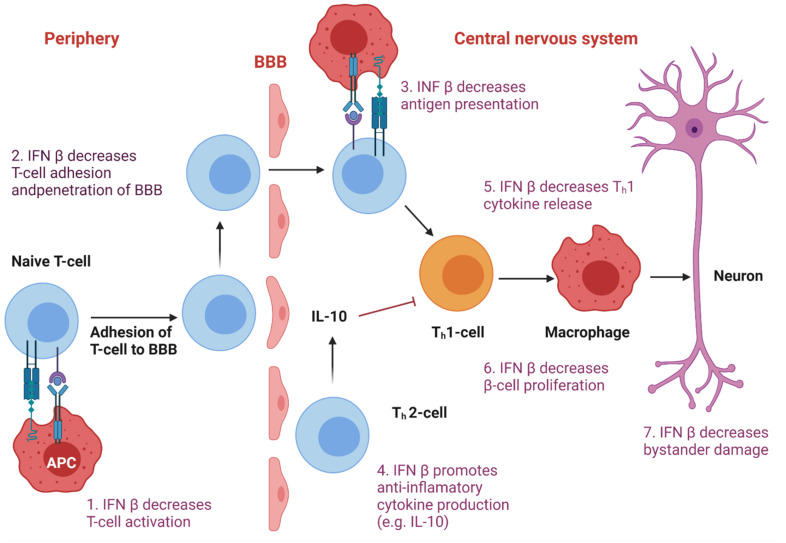
Biological activity of IFNβ in autoimmune diseases. Its action directly increases the expression and concentration of anti-inflammatory agents and down-regulates the expression of proinflammatory cytokines. The same pathway that enables interferon beta’s biological effects mediates its mechanism of action in MS. This cytokine binds to its specific receptors IFNR I and IFNRII on the surface of the main cells of the immune system (DC, TH1 TH2, and B cells). Ligand-receptor binding triggers a cascade of events within these cells that results in positive feedback from the molecule, increasing IFNβ levels and producing the expression of multiple ISGs such as MHC Class I, Mx protein, 2′/5′-oligoadenylate synthetase (OAS), β2-microglobulin and neopterin. These products have been found in serum and cellular fractions of blood from patients treated with interferon-beta. Created with BioRender.com.

## Data Availability

Not applicable.

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
