# Peer review of "Forms and Methods for Interferon’s Encapsulation"

_pharmaceutics, 2021, doi:10.3390/pharmaceutics13101533_

Round 1
Reviewer 1 Report
The authors reviewed the introduction and effectiveness of a continuous and efficient transport system through nano-encapsulation of interferons (IFNs) that inhibit the growth and replication of viruses and cancers in the body.
1. Nanoencapsulation
“This configuration allowed the transport of the molecule into the liver without 420 passing through the RES, maintaining a stable plasma concentration for 24 hours”
Authors should explain the process by which plasma concentrations are maintained as changes over time.
2. Microencapsulation
The authors have categorized and summarized various encapsulation methods for INF in a table, but advantages and disadvantages should be added.
In the case of IFNγ, it was mentioned that the study on microencapsulation was almost stopped. Why are recent results on IFNγ encapsulation not available in the literature?
It would be great if you could add a reference to the cons and side effects of IFNß.
3. Some minor typos, grammar and syntax errors should be carefully revised and corrected accordingly.
IFN β decreases T-cell adhesion andpenetration of BBB
IFN β decreases T-cell adhesion and penetration of BBB
IFN β promotes anti-inflamatory cytokine production (e.g. IL-10)
IFN β promotes anti-inflammatory cytokine production (e.g. IL-10)
Author Response
We appreciate the time and efforts by the editor and referees in reviewing this manuscript. We have addressed all issues indicated in the review report, and believe that the revised version can meet the journal publication requirements. We have responded specifically to each suggestion below. To make the changes easier to identify where necessary, reviewer’s comments have been numbered.
REVIEWER 1
The authors reviewed the introduction and effectiveness of a continuous and efficient transport system through nano-encapsulation of interferons (IFNs) that inhibit the growth and replication of viruses and cancers in the body.
Comment 1: NANOENCAPSULATION
Comment 1a:"This configuration allowed the transport of the molecule into the liver without 420 passing through the RES, maintaining a stable plasma concentration for 24 hours."
The authors should explain the process by which plasma concentrations are maintained as changes over time.
Response 1a:
Line 455
The main route of elimination of IFNα is renal catabolism, with minimal amounts of IFNα in the urine, while hepatic metabolism and biliary excretion are minor routes of elimination. In 2011, Giri and coworkers obtained PLGA nanoparticles adsorbed with hepatitis B virus surface antigens. This configuration allowed the transport of the molecule into the liver without passing through the RES, maintaining a stable plasma concentration for 24 hours. The distribution and opsonization of PLGA nanoparticles appeared to be influenced by size (174 nm) and surface characteristics with uniform distribution. When using Intravenous administration, the filtering effect of the pulmonary capillary bed removed large particles and aggregates. HBsAg-modified particles remained in the circulation longer than the uncoated counterpart, with a high blood concentration of the loaded drug for a prolonged period, with a targeting effect.
Comment 2: MICROENCAPSULATION
Comment 2a: The authors have categorized and summarized various encapsulation methods for INF in a table, but advantages and disadvantages should be added.
Response 2a:
A column with advantages/disadvantages was added for each formulation.
According to the systems, it was added in the text (see Table 1) so that the reader is oriented that all forms of encapsulation are in the table. The table was placed before discussion and results.
Comment 2b: In the case of IFNγ, it was mentioned that the study on microencapsulation was almost stopped. Why are recent results on IFNγ encapsulation not available in the literature?
Response 2b:
Line 427
The problems reported by several authors regarding the protein's stability for this encapsulation system led to the evaluation of alternative strategies such as the formulation in nanoparticles. Further and more recent results for the encapsulation of this subtype are not yet available in the literature.
Comment 2c: It would be great if you could add a reference to the cons and side effects of IFNß.
Response 2c:
On line 166, there is cite [33], which corresponds to the cons and side effects of IFNß.
Comment 3: Some minor typos, grammar, and syntax errors should be carefully revised and corrected accordingly.
- IFN β decreases T-cell adhesion and penetration of BBB
- IFN β promotes anti-inflammatory cytokine production (e.g., IL-10)
Response 3: In figure 7, points 2 and 4 were corrected.

Reviewer 2 Report
The review is informative and well written. I only see few points to correct:
Major points:
The structures of the PEG used for IFN PEGylation should be presented as should be discussed the influence on the number of PEG chains and their lenght on stability.
Alternatives to PEGylation such as PASylatin, PCB, XTEN should be presented.
Authors need to reproduce critical figures and results from papers.
Methods of conjugation and encapsulation should be explained with schemes.
Authors should moderate their conclusions on the failure of liposomes-based systems. Reviews on approved liposome therapeutics coud be cited, such as DOI 10.1002/btm2.10003
The RES term is outdated, this should be replaced by RES (Reticulo Endothelial System) along with appropriate references.
A Table with clinically approved products and majr clinical trials should be added.
Information on funding should be added.
Minor points:
In the image Abstract "pegulation" should be "pegylation".
Author Response
We appreciate the time and efforts by the editor and referees in reviewing this manuscript. We have addressed all issues indicated in the review report, and believe that the revised version can meet the journal publication requirements. We have responded specifically to each suggestion below. To make the changes easier to identify where necessary, reviewer’s comments have been numbered.
Comment 1: The structures of the PEG used for IFN PEGylation should be presented as should be discussed the influence on the number of PEG chains and their lenght on stability.
Response 1:
Line 220
PEGylation was the first formulation aimed at improving the pharmacological properties of IFNs. It consists of the covalent bonding of poly(ethylene glycol) (PEG) chains to a drug to increase its half-life, reduce its clearance, and improve its pharmacokinetics and pharmacodynamics. PEG is FDA approved for systemic applications due to its pharmaceutically relevant properties: increases IFN solubility and stability by decreasing proteolytic degradation; reduces renal clearance rate by increasing the size of the renal boundary molecule, decreases plasma clearance, improves the safety profile of the protein by protecting antigenic and immunogenic epitopes; and increases circulation time, high mobility solutions, and low toxicity
Line 228
Through covalent conjugation of PEG to IFN molecules, several formulations of PEGylated IFNs were developed using two types of conformation, linear and branched: Pegasys® (Hoffmann La Roche Inc.), conjugate IFNα-2a to a 40 kDa branched-chain via amide linkages; PegIntron® and ViraferonPeg® (Schering-Plough Corporation), link IFNα-2b to a 12 kDa linear molecule via a urethane linkage; and Plegridy® (Biogen Inc.), couple IFNß-1a to a 40 kDa linear PEG chain with amide linkages. PEG conjugation does not alter protein conformation significantly, but several aspects such as chain size and structure influence biological activity. Increasing molecular weight increases the half-life, and coupling 30-40 kDa PEGs achieves this effect. IFNs that bind to branched chains has a lower biological activity loss than those conjugated to linear chains because binding to different amino acids binds a more excellent polymer. The half-life and stability in branched PEGylations are higher because this structure decreases the glomerular filtration of proteins and protects their surface to a greater extent. These forms of PEGylation with IFNs showed reduced excretion through the kidneys, with a five- to tenfold increase in half-life time, resulting in more stable drug concentrations in the plasma of patients and replacing systemically applied IFNs. This platform directly enhanced the drug's pharmacokinetics, making possible less frequent dosing intervals of PEG-IFNα on patients with chronic hepatitis B and C while still effectively reducing their viral load. PEGylation of IFNß used in multiple sclerosis therapy resulted in a more comfortable regimen by reducing the dosage. Similarly, PEG- IFNγ conjugation has been evaluated, finding an increase in drug half-life of up to 32-fold in in vivo studies. Currently, some formulations of PEGylated IFNs are in preclinical and clinical trial stages, including PEGylated IFNβ (preclinical testing completed) and PEGylated IFNα (in preclinical studies), both from Bolder BioTechnology, as well as RogPEGinterferonα-2b (P1101) from PharmaEssentia (in preclinical studies).
Line 256
Other similar forms of conjugation have been evaluated: PASylation as an alternative to PEGylation, to increase the half-life of IFNs, which consists of binding by genetic fusion of cytokines with a polypeptide sequence rich in proline, alanine, and serine (PAS). Several investigations have shown that PASylation of IFNα and IFNß increases their stability, bioavailability, and biological activity, with minimal toxicity and immunogenicity. Despite the scientific evidence gathered, none of these formulations have reached the clinical trial stage.
Comment 2: Methods of conjugation and encapsulation should be explained with schemes.
Response 2:
See Figure 4
Comment 3: Authors should moderate their conclusions on the failure of liposome-based systems. Reviews on approved liposome therapeutics could be cited, such as DOI 10.1002/btm2.10003
Response 3:
Attached reference was considered
Line 276
There are different liposomal formulations for encapsulating chemotherapeutic drugs, antifungals, and vaccines, currently approved by regulatory agencies for clinical application
Line 310
Despite the success that this form of encapsulation has had with different drugs, numerous challenges affect the effectiveness of liposomes in formulations with IFNs. Hypersensitivity, opsonization, uptake by the Reticulo Endothelial System (RES), and immunosuppression are the primary negative responses of the immune system to liposomes. It is worth noticing that the production of lipid-based transport systems is expensive and that liposomes are not very stable because of their susceptibility to fusion, aggregation, or assembly without these cytokines. For these reasons, there are currently no liposomal systems that encapsulate IFNs approved by regulatory agencies for clinical application.
Comment 4: The RES term is outdated, this should be replaced by RES (Reticulo Endothelial System) and appropriate references.
Response 4:
Line 161
IFNs’ therapy has encountered difficulties due to the size of the molecules, their sensitivity to degradation, and rapid elimination from the blood circulation.
The term was updated in other mentions
Comment 5: A Table with clinically approved products and major clinical trials should be added.
Response 5:
|
Name (company) |
Particle type/drug |
Investigated application/indication |
ClinicalTrials.gov/identifier (phase) |
Year |
|
Biolex Therapeutics, Inc
|
Poly(ether-ester) microspheres (Poly-Active) |
Hepatitis C, Chronic Locteron |
Phase 2B NCT00863239 Phase 2B NCT00953589 |
2012 |
Each formulation was exhaustively reviewed, and Table 1 refers to the last in vitro or in vivo study conducted with that encapsulation for interferons. For this reason, it is not possible to add a table with the approved products because most of these formulations have had an investigative nature and have not reached preclinical or clinical stages, except for Locteron, which does appear with two phase 2 clinical trials on the site https://clinicaltrials.gov/.
Comment 6: Information on funding should be added.
Response 6:
Nanoformulations as a controlled release system. INNBIO-UdeC 032020.
Comment 7: Minor points: In the image Abstract, "pegulation" should be "pegylation".
Response 7:
Abstract figure corrected

Reviewer 3 Report
Comments in the attached file

Author Response
We appreciate the time and efforts by the editor and referees in reviewing this manuscript. We have addressed all issues indicated in the review report, and believe that the revised version can meet the journal publication requirements. We have responded specifically to each suggestion below. To make the changes easier to identify where necessary, reviewer’s comments have been numbered.
The manuscript by Ramos and co-authors presents various delivery methods for INFs, including micro and nanoparticles, liposomes, and polymeric micelles. Encapsulation in nanovectors with optimized structural and biopharmaceutical properties is an open challenge in the case of INFs. The review is quite complete, summarizing formulations presented from 1996 to 2021. The review can be published after revision.
Main points:
Comment 1: The overall layout of the manuscript is quite confusing
Response 1:
The table was moved to the end of the encapsulations section so that the reader understands that it refers to all encapsulation forms presented in the manuscript.
Line 503
The current review showcases an updated summary of all forms of encapsulation including liposomes, micelles, microparticles and nanoparticles for IFNα, IFNß, and IFNγ cytokines from 1996 to March 2021, considering the following parameters: encapsulating matrix, route of administration, encapsulation method, physical properties, target, advantages, and disadvantages of each formulation. Table 1 summarizes these systems.
Comment 2: Are IFNs loaded in the hydrophilic or hydrophobic regions of nanovectors (liposomes, micelles …)? What is the size of IFN? The encapsulation of molecules depends on parameters as their charge, steric hindrance, hydrophobicity. A section describing the main physico-chemical features of IFNs is necessary to better understand the significant parameters affecting the loading potential, the stability, the release profile of the different delivery vectors, thus leading the design of new formulations.
Response 2:
Line 181
IFNs in their natural low molecular weight form (~20 kDa) are glycosylated proteins, but this post-translational modification does not play a functional role. Obtaining IFNs was initially derived from leukocytes and lymphoblastoid lines, but extraction of proteins from natural producers suffers from limitations that limit regular large-scale production. Recombinant DNA technology became an excellent option to produce these therapeutic proteins. Recombinant IFNs are mostly non-glycosylated with identical biological activity to their natural counterparts. They possess similar structures adopting a unique α-helix topology relative to other proteins. These molecules possess amphipathic characteristics, with hydrophobic and hydrophilic regions conferring solubility. Their instability, molecular size, hydrophilicity, low permeability, rapid clearance from circulation, and high susceptibility to degradation at low pH and in the presence of proteases have restricted their therapeutic application. Formulations that protect IFNs from degradation and achieve prolonged releases with adequate biological activity are required. Drug design systems that encapsulate therapeutic proteins maximize their biological potential, provide transport matrices that avoid the influence of weak non-covalent interactions (van der Waals forces) and electrostatic interactions that alter protein stability. It also protects the cargo proteins from degradation by enzymes found at the administration site or during transport to the site of action, thus increasing their half-life.
Line 197
New encapsulated formulations for IFNs have demonstrated several challenges, such as electrostatic interactions between IFNs (isoelectric point) and acidic end-groups of the encapsulation matrices (hydrolysis) with consequences on release; the pH of the formulation buffer and its variants with impact on solubility, stability, and aggregation. Efficient encapsulation has been related to the use of stabilizers that support particle size modulation and correlate with release patterns and biological activity. Some of the encapsulations explored have shown the necessity to consider the polarity of the protein with respect to the encapsulant. For example, in amphipathic nanovectors (polymeric micelles), molecules are encapsulated by stimulating protein-like polarity so that hydrophobic structures interact with hydrophobic parts of the system, and hydrophilic portions interact with hydrophilic regions. Protein aggregation is related to hydrophobic interactions, encapsulating it through hydrophilicity to ensure stability (see Table 1).
Line 208
Some IFNs' transport systems that have been studied and evaluated include PEGylation, self-assembled nanostructures such as liposomes and micellar systems, microparticles, and nanoparticles (metallic, polymeric, or hybrid) (Fig. 4). For developing these platforms, criteria of safety, biocompatibility, biodegradability, and compatibility of the encapsulating material with the drug must be considered and comply with the parameters that determine the functionality of a nanoparticle, such as its size, shape, and surface characteristics. However, bringing this approach towards clinical application requires careful evaluation of efficacy, safety, and manufacturing.
Comment 3: Fig. 1 . Left panel. The representation of the induction of type I IFN could be improved. TLR is not linked to the cascade, MAVS are not defined.
Response 3:
The TLR was connected to the cascade and the MAVS were defined.
Comment 4: Line 171. self-assembled nanostructures. Liposomes and micelles are self-assembled nanostructures too. Authors have to identify this class of nanovectors better
Response 4:
Line 208
Some IFNs' transport systems that have been studied and evaluated include PEGylation, self-assembled nanostructures such as liposomes and micellar systems, microparticles, and nanoparticles (metallic, polymeric, or hybrid) (Fig. 4).
Comment 5: Line 194. This platform directly enhanced the drug's pharmacokinetics, making possible a reduction in the amount of PEG-IFNα injections on patients with chronic hepatitis B and C while still effectively reducing their viral load. Line 201. PEGylation causes loss of IFN activity (up to 80% of native IFN), which leads to an increase in doses, and thus a more excellent induction of toxicity.
Authors have better to describe the effect of pegylation on the dosage.
Response 5:
Line 241
This platform directly enhanced the drug's pharmacokinetics, making possible less frequent dosing intervals of PEG-IFNα on patients with chronic hepatitis B and C while still effectively reducing their viral load
Line 250
The loss of IFN activity caused by PEGylation is up to 80% of native IFN, which increases the amount of drug necessary to obtain an antiviral effect equivalent to that of native cytokine, and thus a more significant induction of toxicity. Therapy with these encapsulated formulations can cause a range of adverse events, from mild to severe, such as diabetes, liver neoplasms, or psychotic disorders. The decrease of bioactivity could not always increase the in vivo therapeutic efficacy of IFN, so treatments with these molecules are often unsatisfactory and should be discontinued.
Comment 6: Line 264. Polymeric micelles are macromolecular assemblies formed from blocks of copolymers. They possess a two-phase structure: a spherical inner core to encapsulate desired molecules; an outer layer or corona that determines hydrophobicity, charge and allows modifications to their surface
Polymeric micelles are colloids formed in solution by self-assembling of amphiphilic polymers. The outer corona mainly exposes hydrophilic regions to the aqueous solvent. What do the authors mean with "blocks of copolymers" and with "corona that determines hydrophobicity"?
Often the amphiphilic molecules are block copolymers, a type of polymer in which each molecule consists of two or more simple different polymers (blocks) joined in some arrangement. Block copolymers with two, three, and more blocks are called diblocks, triblocks, and multiblocks.
A more accurate introduction of the structure of polymeric micelles and on their advantages for drug delivery is required.
Response 6:
Line 318
Polymeric micelles are nanocarriers formed by the spontaneous arrangement of amphiphilic block copolymers in aqueous solutions. Block copolymers are macromolecules of two or more different polymers joined by covalent bonds to form one structure. Its molecular conformation depends upon the number of blocks. Diblock copolymer consists of two homopolymers, while triblock copolymer has three homopolymers. More complicated architectures such as mixed arm block copolymers contain more than three polymer chains covalently joined at a common branching point. Polymeric micelles possess a two-phase structure: a hydrophobic core and a hydrophilic corona that allows modifications to their surface. Polymeric micelles have several advantages for drug delivery, such as their increased solubility, enhanced stability of the molecule, structural flexibility, capacity to encapsulate a wide range of therapeutics, and the possibility of adjusting their size at the nanometer scale. Modifications in the corona make it possible to reduce their clearance by the RES, thus prolonging their circulation time. In this way, it is feasible to decrease the drug dose and the toxicity associated with drugs such as IFNs.
Comment 7: Line 288. However, this process involves altering its genetics by changes in the sequences of the copolymer blocks [86] so that the process becomes more complex compared to other encapsulation strategies.
I don't understand why the change in the sequence of copolymer is defined a genetic change and why this process is complex.
Response 7:
Line 342
Particle stability was another problem, which they resolved by regulating the assembly with temperature changes. However, this process involves a genetic fusion of IFNα with the polypeptide block copolymer sequence, so that the process includes extra steps compared to other encapsulation strategies.
Comment 8: Line 307. The current review showcases an updated summary of all forms of micro and nano- encapsulation for IFNα, IFNß, and IFNγ cytokines from 1996 to March 2021, considering the following parameters: encapsulating matrix, route of administration, encapsulation method, physical properties, and formulation target. Table 1 summarizes these systems.
Liposomes and polymeric micelles are listed in Table 1, although they have been presented in the previous sections. The overall layout of the manuscript is quite confusing
Response 8:
The table was moved to the end of the encapsulations section so that the reader understands that it refers to all encapsulation forms presented in the manuscript.
Comment 9: Minor
- Line 19 (with). in
- Line 21 (with currently active) is currently active with Line 28 (for these cytokines INF) for INF
- Line 70 (This) this
- Line 176 However is not appropriate
- Line 241 In 2012, Li et al. encapsulated IFN in liposomes with 83.5 nm cyclic peptides. Incorrect sentence Line 438 Although in vitro studies and in hepatocytes of Wistar rats administered with the formulation showed no evidence of toxicity, they observed mild side effects in the kidney, such as whiteness and pyelectasis. The formulation of the sentence needs to be improved
- Line 453. I think this paragraph does not refer to IFNs, but it is a general comment
- Line 466 ( Discussion and conclusion) Discussion
Response 9:
All of these were reviewed and corrected.

Round 2
Reviewer 1 Report
Authors revised the manuscript as per the reviewer’s comments, now it can be considered for publication in Pharmaceutics.
Reviewer 3 Report
The manuscript can be accepted in the present form